# Effects of Supervised Physical Exercise as Prehabilitation on Body Composition, Functional Capacity and Quality of Life in Bariatric Surgery Candidates: A Systematic Review and Meta-Analysis

**DOI:** 10.3390/jcm11175091

**Published:** 2022-08-30

**Authors:** Andrea Herrera-Santelices, Graciela Argüello-Florencio, Greice Westphal, Nelson Nardo Junior, Antonio Roberto Zamunér

**Affiliations:** 1Facultad de Ciencias de la Salud, Universidad Católica de Maule, Talca 3480112, Chile; 2Servicio de Medicina Física y Rehabilitación, Hospital San Juan de Dios, Curicó 3343005, Chile; 3Department of Health Sciences, Nutrition and Dietetics Career, Faculty of Medicine, Pontificia Universidad Católica de Chile, Santiago 8331150, Chile; 4Multidisciplinary Center of Obesity Studies, Department of Physical Education, Sate University of Maringa, Maringá 87020-900, Brazil; 5Laboratory of Clinical Research in Kinesiology, Department of Kinesiology, Universidad Católica del Maule, Talca 3480112, Chile

**Keywords:** prehabilitation, bariatric surgery, obesity, physical exercise, quality of life

## Abstract

Background: Prehabilitation is a strategy used aiming to reduce the risk factors and complications of surgery procedures, but there is no consensus on the effectiveness of supervised physical exercise and its optimal prescription during this phase. Objectives: To determine the effects of exercise prehabilitation on body composition, functional capacity and quality of life in candidates for bariatric surgery. Search methods: A search was conducted in PubMed, Web of Science, SciELO, Scopus, MEDLINE and CINAHL. Selection criteria: Only randomized clinical trials that examined the effectiveness of supervised physical exercise were included. The main outcomes were body composition, functional capacity, quality of life and surgical outcomes. Data collection and analysis: Two researchers independently selected the literature, extracted the data and evaluated the risk of bias. A third researcher was consulted when a consensus was not reached. The risk of bias was assessed by the tool recommended by the Cochrane Collaboration, the quality of the evidence by GRADE, and to analyze the effects of prehabilitation on the primary objectives, RevMan software, version 5.3 was used. Main results: The search resulted in 4550 articles, of which 22 met the eligibility criteria, leaving 5 articles selected for this review. One article was assessed as a high bias risk and four as an uncertain risk, which included 139 candidates for bariatric surgery. Most of the studies evaluated the body composition, functional capacity and quality of life; none reported surgical outcomes. Conclusions: Supervised physical exercise has positive effects on the body composition, functional capacity and quality of life; there was no evidence for surgical outcomes, which opens up a field of study for future research of this population.

## 1. Introduction

Obesity is a chronic disease that is progressive, recurrent and creates health problems, depending on the topographical location of excessive fat deposits. The most common health problems are metabolic syndrome, high blood pressure, sarcopenia, osteopenia, diabetes mellitus, obstructive sleep apnea syndrome, dyslipidemia, depression and anxiety disorder, among others [1]. Therefore, several treatment pharmacological, non-pharmacological and surgical strategies have been proposed [2]. In this sense, bariatric surgery (BS) has proven to be effective in solving comorbidities and promoting long-term weight loss in people with obesity [3].

When a patient undergoes major abdominal surgery, such as BS, evidence suggests that the proper preparation decreases the risks associated with the surgical procedure and promotes a recovery that includes a higher pain tolerance, fewer hospital stays, less need for rehospitalization and less surgical complications in the short and long term (e.g., venous thrombosis, surgical wound dehiscence, bowel obstruction and adhesions development, among others) [4,5,6].

Enhanced Recovery After Surgery (ERAS) is the current protocol used by different surgical specialties to promote post-op recovery [7]. Regarding BS, the ERAS protocol recommends several procedures and lifestyle changes, including diet control and the use of some specific pharmacological prescription [7]. In addition, patients are encouraged to participate in a preoperative weight loss program [6]. In that regard, it is well-known that exercise has multiple benefits for a person’s physical and mental health, especially for those with obesity. Exercise has been used as one of the main strategies for weight control and in the treatment of different associated comorbidities, contributing to glycemic control; lowering resting blood pressure and improving body composition, cardiorespiratory fitness, sleep quality and quality of life [8,9]. Therefore, regular physical exercise could have an important role in wight loss programs during prehabilitation for BS. However, despite some randomized controlled trials (RCTs) having addressed this subject [10,11,12], there is currently no clarity regarding its effectiveness on surgical outcomes (e.g., hospitalization days, post-op pain tolerance, short- and long-term complications, rehospitalization, etc.); mortality and other indices to support its recommendation in the ERAS protocol for BS [13].

Therefore, the objective of this systematic review is to determine the effect of prehabilitation on the body composition, functional capacity, quality of life and surgical outcomes in patients who are candidates for BS.

## 2. Materials and Methods

The protocol for this systematic review was registered on PROSPERO; the registered number is: CRD42021261474 [14]. This systematic review was conducted in accordance with the PRISMA guidelines.

### 2.1. Search Plan and Literature Selection

The articles were searched in the following electronic databases: PubMed, Web of Science, SciELO, Scopus, MEDLINE and CINAHL between 1 and 31 July 2021 without restrictions of language or publication date. The descriptors used were: “Prehabilitation”, “Physical Exercise”, “Body Composition”, “Functional Capacity”, “Quality of Life”, “Surgical Outcomes” and “Bariatric Surgery”; for the combinations of these, the Boolean operators “AND” and/or “OR” were used. All studies were exported to the 5.4 version of StArt (State of the Art through Systematic Review) software (developed by the Federal University of São Carlos).

### 2.2. Types of Participants/Population

Inclusion criteria

Eighteen-year-old adults or older, both sexes and candidates for a first bariatric surgery who were included in a prehabilitation program were included.

Exclusion criteria

Candidates for a second BS or reconversion surgery were excluded.

### 2.3. Types of Intervention/Exposition

For this review were considered randomized controlled clinical trials that applied supervised physical exercise programs described as aerobic exercise training, resistance exercise training or included both, with a duration of at least one week and performed before bariatric surgery.

### 2.4. Types of Comparator/Control

The control was considered as a group receiving no intervention or only the standard care, defined as advice, counseling, brochures or leaflets on various health topics or educational intervention of any kind.

### 2.5. Types of Outcome/Results Measurements

The studies were included if they reported the effect of the intervention on one or more of the following outcomes: (1) Body composition: evaluated through dual-energy x-energy absorptiometry or bioelectrical impedance; (2) Functional capacity: evaluated through a functional test, for example, the six-minute walk test (6MWT), the direct or indirect maximum rate of oxygen consumption (VO_2_ max), the sitting to standing test or the step test among others; (3) Quality of life: evaluated through any quality of life questionnaire and (4) Surgical outcomes such as the number of hospital stay days, the need for rehospitalization within 30 days and post-op complications at 30 days were initially considered. However, they were disregarded, since these outcomes were not reported by any of the included studies.

### 2.6. Studies Selection

Two researchers (AH and GA) examined, independently, the studies that were identified by the search strategy using the 5.4 version of the StArt software. They were firstly identified by reading the title and the abstract. In order to be selected, the abstracts had to clearly identify the studies’ design, population, intervention and outcome measurements, as previously described. In the event of a disagreement, a third researcher (GW) was consulted who determined whether the article was included or not. Then, both researchers moved on to reading the entirety of the potentially eligible articles and examined their eligibility according to the inclusion criteria. Again, if there was a disagreement, a third researcher was consulted. Conference reports and letters to the editor were excluded.

### 2.7. Data Extraction

After selecting the studies, two researchers (AH and GA) independently extracted the data according to a “standard data extraction form” created by two researchers from the team (AZ and AH). In the event of a disagreement between the reviewers, the data was subjected to consensus or arbitration by a third reviewer (GW). Both reviewers conducted a pilot test of data extraction using the standard data extraction form on two randomized controlled clinical trials that were related to exercise and its effect on cardiovascular risk factors.

### 2.8. Risk of Bias Assessment

The studies’ risk of bias was independently evaluated by two researchers (AH and GA) through the “Cochrane Risk of Bias Tool”, with 6 bias domains: selection, realization, detection, attrition, report and others. Each domain is qualified as a high, low or uncertain risk of bias (https://www.bmj.com/content/343/bmj.d5928, accessed on 25 October 2021). Punctuation disagreements were discussed between them until a consensus was reached; in case there was no agreement, a third researcher (GW) was consulted.

### 2.9. Evaluation of the Quality of Scientific Evidence

The quality of evidence in the studies was evaluated under GRADE (Grading of Recommendations Assessment, Development and Evaluation) criteria, including the study’s limitations, the consistency of the effect, the inaccuracy, the evidence, and the publication bias. The webpage www.gradepro.org was used.

### 2.10. Statistical Analysis

A meta-analysis was carried out using Review Manager software (RevMan, version 5.3). Continuous outcomes were meta-analyzed using a random effects model and standard mean differences (SMDs). Heterogeneity was quantified by the I-squared (I^2^) test and classified as low: I^2^ < 25%, moderate: I^2^ = 25.1–50% and high: I^2^ > 50.1%.

## 3. Results

### 3.1. Article Selection

Figure 1 shows the flow chart pertaining to the identification of the studies and the selection process of these. The results from searching the database were 4550 articles, of which 626 were extracted while screening, because they were duplicates. Thus, 3924 were analyzed by reading the title and its abstract, excluding 3902; then, 22 met the eligibility criteria, 17 were excluded when the entire article was read and, finally, 5 studies were included in this review.

### 3.2. Articles’ Descriptions

Table 1 shows, in detail, the description of each article. The five articles selected for review and meta-analysis were published between 2010 and 2021 in English; four of them were performed on the American continent [11,15,16,17] and one in Europe [18].

### 3.3. Participants

A total of 139 participants were enrolled in the five selected studies. The data from 115 participants were used for the meta-analysis on the body composition (BMI) [11,15,17,18], 75 participants for the fat mass percentage (FM%) [11,15,18], 46 for the free fat mass (FFM Kg) [15,18], 61 participants for the meta-analysis of the 6MWT [17,18] and 53 participants for the meta-analysis of the quality of life total score [15,16,18]. Regarding the demographics characteristics of the included studies, the sample size varied between 7 and 57 participants, the age ranged between 28 and 54 years old, 116 participants were women and all the studies included men.

### 3.4. Types of Intervention/Exposition

The duration of the intervention programs in the included studies ranged from 8 to 16 weeks. Twelve weeks of intervention were used in three out of the five articles [11,15,16]. One study had an intervention session frequency of three times a week [15] and twice a week. The duration of each session varied between 25 and 80 min, and the location was either a hospital gym or an educational institution where the researchers belonged [11,15,16,17,18].

In regards to the type of training, the combination of aerobic and resistance exercise was used in three studies; the other two used aerobic [11] and resistance [18] training separately. The average amount of sessions was 26.4, with a range between 16 and 32 sessions.

### 3.5. Types of Comparator/Control

Only one study did not use any kind of intervention as a control [16]. Standard care was used for the rest of the studies, counseling being the one used most. One study added cognitive–behavioral therapy to the standard care [17].

### 3.6. Risk of Bias Evaluation

Figure 2 and Figure 3 show detailed results of the general risk of bias evaluation and the evaluation per study, respectively. The randomization generation sequence (selection bias) was judged as a low risk in all the included studies. On the other hand, the selective reporting data (report bias) was classified as an uncertain risk in all the studies. A high risk of bias can be noted in 20% of the included articles for the following items: incomplete results data (attrition bias), blinding of the participants and personnel (performance bias) and for other biases.

### 3.7. Prehabilitation Effects on the Outcome Measurements

#### 3.7.1. Body Composition

Four studies measured the body composition and reported the BMI [11,15,17,18], FM% [11,15,18] or FFM, expressed in kilograms [15,18]. Other indexes were reported, such as abdominal fat in percentage, abdominal muscular mass in kilograms and fat mass in kilograms. However, they were not used for the meta-analysis, since it was only one study [18]. Overall, the results indicated no significant effect of prehabilitation in favor of the experiment or controls for body composition indexes (*p* > 0.05). Figure 4 shows the forest plot for BMI using the random effects model to compare the experimental versus control groups. The results showed a pooled effect of −0.71 (IC_95%_: −1.55 to 0.1; *p* = 0.09). The heterogeneity was 76%, and the quality of evidence was very low (Table 2).

Regarding the effect of an intervention on the FM% (Figure 5), the three studies included in the analysis resulted in a pooled effect of 0.38 (CI_95%_: −0.08 to 0.84; *p* = 0.11). The heterogeneity was 0%, and the quality of the evidence was moderate (Table 2).

Figure 6 shows the analysis of the FFM kg subgroup. The pooled effect size was −0.41 (IC95%: −1.00 to 0.18; *p* = 0.17) and a heterogeneity of 0%, with a moderate quality of evidence (Table 2).

#### 3.7.2. Functional Capacity

All studies evaluated the functional capacity. The 6MWT was the most used test. Two studies reported the results as the distance traveled in meters [17,18] and one in the number of total steps [16]. Baillot studies [11,15] did not show the values for the test results, which is why they were not included in the meta-analysis.

The VO_2_ max was an outcome reported for only one study, which was estimated from a 6MWT equation [17]. Other indicators were used for the outcome report: the chair stand test [11,15,17,18], postural stability test, abdominal strength, core flexor strength and the modified push-up test were not analyzed statistically.

Figure 7 and Table 3 show the random effect analysis of the functional capacity for the 6MWT and the quality of evidence, respectively. The pooled effect was 2.59 (IC_95%_:1.89–3.30; *p* < 0.0001) in favor of exercise, showing a low heterogeneity (I^2^ = 0%) and high quality of evidence.

#### 3.7.3. Quality of Life

Four studies evaluated the quality of life [11,15,16,18]: the SF-36 questionnaire, Weight-Related Quality of Life (WRQOL) and Obesity Specific Quality of Life (OSQOL) were used. Baillot et al. (2016) [11] did not report the post-intervention values, so it was not included in the meta-analysis. The SMD was used to combine the results of the three included studies. The random effect model resulted in a pooled effect size of 0.88 (CI_95%_: 0.23–1.99; *p* = 0.12; Figure 8), and the quality of evidence was moderate (Table 4).

#### 3.7.4. Surgical Outcomes

No study reported the results for this outcome measurement.

## 4. Discussion

This systematic review’s objective was to determine the effect of prehabilitation on the body composition, functional capacity, quality of life and surgical outcomes in patients who are candidates for bariatric surgery. In the last 10 years, a series of studies have evaluated the effects of physical training programs in the context of BS; most of which were done after the surgery. To the best of our knowledge, only two reviews [19,20] reported the effects of exercise on BS candidates in some variables considered in this study, but they did not perform a meta-analysis. Moreover, only two studies included in the previous reviews were RCTs, strengthening the relevance of the present study.

The results of this systematic review of RCTs showed that supervised exercise as prehabilitation before BS has positive effects on the body composition (i.e., BMI, FM% and FFM Kg); functional capacity (6MWT) and quality of life. In this sense, our results corroborate the findings of previous systematic reviews on this subject [19,20], who reported, in a descriptive manner, similar results.

The international guidelines for the current treatment recommend that exercise programs for weight loss in obesity prioritize continuous aerobic exercise with a moderate intensity and complement this approach, whenever possible, with resistance training [21]. Although these recommendations are for people who are in nonsurgical treatment for obesity, aerobic exercise was the mostly used intervention modality in the included studies. Three studies combined aerobic and resistance training [11,15,16], one study used only resistance training [18] and the other one used only aerobic exercise [17]. On the other hand, the intensity was heterogenous among the included studies. Regarding the intensity of aerobic exercise, two studies prescribed intensities ranging from 55% to 75/80% of the reserve heart rate [11,15], one study prescribed the exercise intensity ranging from 2 to 4 on the Borg CR10 scale [17] and one did not present details on the exercise intensity [16]. For resistance training, one study prescribed exercise at 50% of one maximal repetition [18], two studies prescribed the resistance intensity according to sex [11,15] and one study did not report details on the intensity prescription [16]. The study that found the greatest improvement on the BMI prior to BS was Marcon et al. (2017) [17], while the greatest improvement in the quality of life was that reported by Arman et al. (2021) [18]. In addition, both studies [17,18] reported significant improvements on the functional capacity. Therefore, considering the protocols are mostly heterogenous among the included studies, it is not possible to conclude what type of training and intensity are the most suitable and effective for BS candidates. Future RCT studies should address this subject to better guide clinicians during prehabilitation.

Regular physical exercise has several effects on metabolism [22]. It is documented that, on obese people, aerobic training at a moderate intensity improves many comorbidity markers associated with it, such as glucose metabolic alteration, dyslipidemia and hypertension, as well as those indicating cardiovascular disease risk factors (e.g., systematic inflammation, oxidative stress and diabetes) [22,23]. Moreover, it also increases free fatty acids oxidation and reduces the total fat and visceral fat [24]. At a muscular level, the increase of the mitochondrial content as an effect from aerobic training at a moderate intensity has a series of metabolic effects (e.g., a higher rate of fatty acid oxidation, a higher breakdown of carbohydrates and a better glucose uptake in the cells, among others), contributing to improving their performance during exercise and, therefore, functional capacity [25]. Those factors could explain the results found in this systematic review.

Regarding the quality of life, the current results corroborated the findings of Carraça et al. (2021) [26]. The authors conducted a systematic review and meta-analysis on the effects of exercise on the quality of life and other psychosocial variables in participants overweight and obese. The results showed that exercise has a positive effect on the quality of life. Regular physical exercise helps in treating depression and anxiety; reduces stress levels, improves sleep quality and has positive effects on the performance of daily life activities, which translates to a better quality of life for people with obesity [26].

## 5. Study Limitations

Although this systematic review and meta-analysis has methodological strengths, some limitations must be mentioned. First, the search for information was performed by only one researcher (AH); however, the terms and search strings were defined by the researchers in collaboration with a university-based librarian with experience in systematic reviews. Second, the fact that there are a limited number of studies that evaluate preoperative interventions can influence the meta-analysis results. Consequently, the results are not conclusive yet. Finally, these results show evidence of the need for studies that include a greater number of participants and other relevant variables such as postoperative complications, days of hospital stay, the need for rehospitalization within 30 days after the surgery, pain tolerance, etc.

## 6. Conclusions

Prehabilitation has positive effects on the body composition, functional capacity and quality of life in patients who are candidates for bariatric surgery. Apparently, supervised aerobic training at a frequency of two times a week and a duration of 45–60 min per session for 12 weeks is the most preferred protocol used for this population. However, there is still a lack of research studying the effects of exercise as a prehabilitation on surgical outcomes.

## Figures and Tables

**Figure 1 jcm-11-05091-f001:**
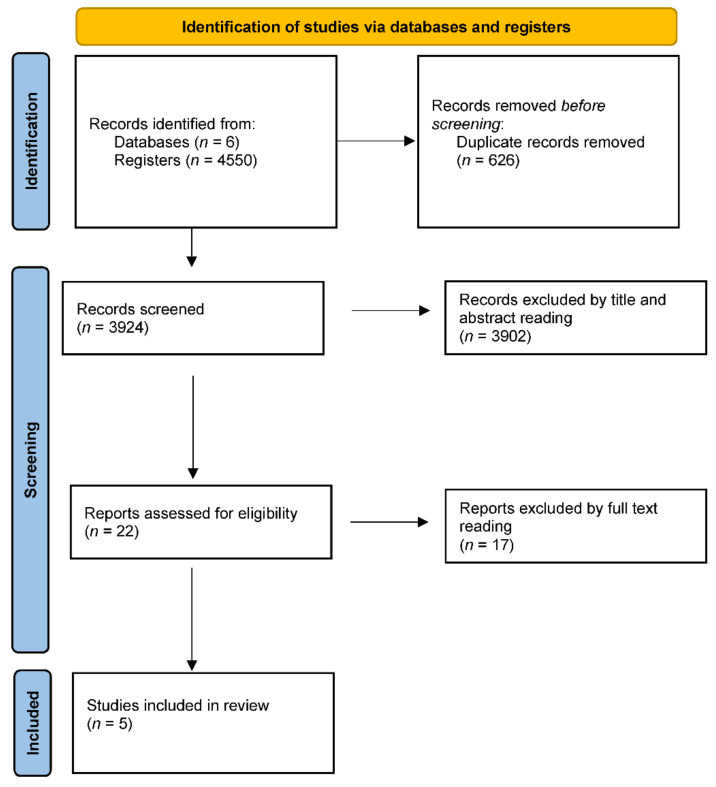
Article identification PRISMA flow chart.

**Figure 2 jcm-11-05091-f002:**
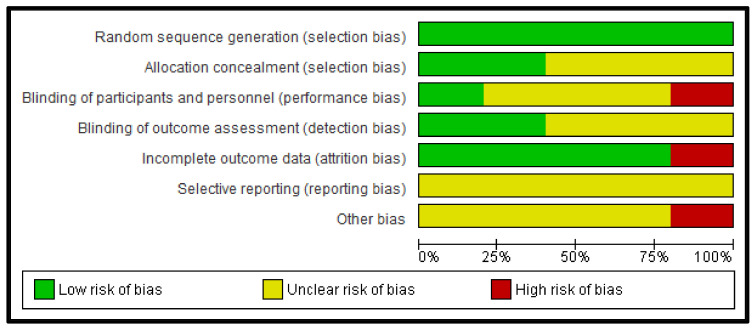
Evaluation of the general risk of bias.

**Figure 3 jcm-11-05091-f003:**
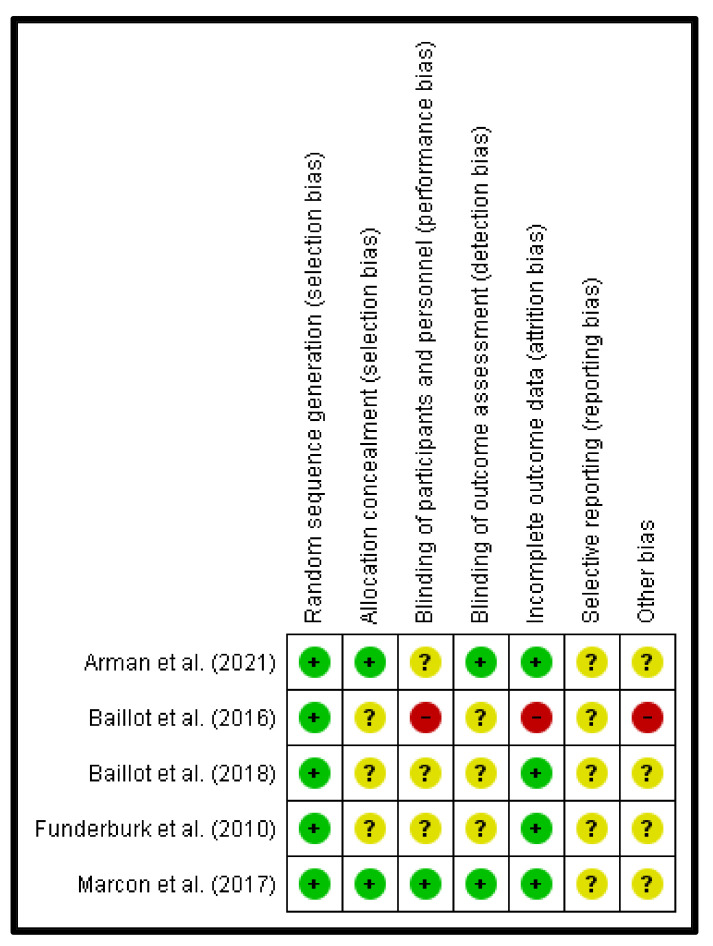
Risk of bias evaluation per study [11,15,16,17,18]. Red (-) = high risk of bias; Yellow (?) = unknown risk of bias; Green (+) = low risk of bias.

**Figure 4 jcm-11-05091-f004:**
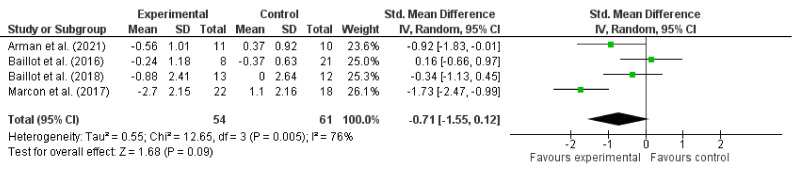
Forest plot of the body composition and the BMI subgroup [11,15,17,18].

**Figure 5 jcm-11-05091-f005:**
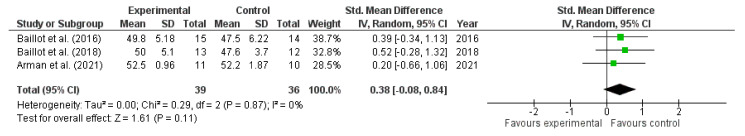
Forest plot of the body composition for the FM% subgroup [11,15,18].

**Figure 6 jcm-11-05091-f006:**
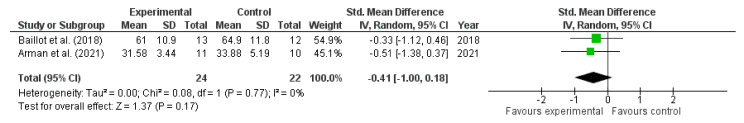
Forest plot of the body composition for the FFM Kg subgroup [11,18].

**Figure 7 jcm-11-05091-f007:**
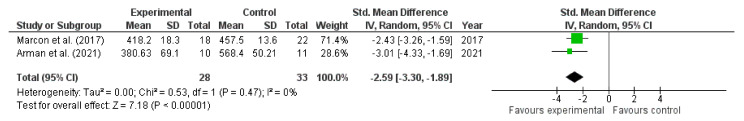
Forest plot of the functional capacity for the 6MWT [17,18].

**Figure 8 jcm-11-05091-f008:**
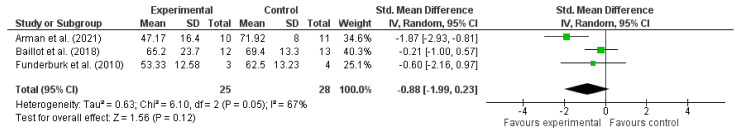
Forest plot of the quality of life total score [15,16,18].

**Table 1 jcm-11-05091-t001:** Description of each article.

[18] Arman et al. (2021)	
Randomization	Software Stratified by Sex and Age Was Used.
**Participants**	21 participants: 1 man—20 women. **Institution:** Rehabilitation and Physical Therapy Department, Health Sciences Department, Istanbul University-Cerrahpasa. **Country:** Turkey. **Inclusion criteria:** candidates to a BS, 18 years or older, both sexes. **Exclusion criteria:** participants with comorbidities that prevent their participation in the prehabilitation program like the existence of acute pain, cardiac pain or a previous dolor heart attack, cardiac failure, diabetes, or uncompensated hypertension.
**Intervention**	**Program of the institution:** 1. Warm-up (10–15 min): walk was performed on a treadmill; heart rate was monitored with a pulse oximeter and as a goal it was set at 50 to 60 heartbeats. 2. Load (30–45 min): exercises for core stabilization were progressively performed in supine position, long sitting position, knee position, crawling position, foot over one leg position and sitting on a ball as exercise. Involved a combination of strengthening, resistance, and balance exercises, along with breathing. Exercises for each main muscle group were performed during 2 cycles of 7 and 10 repetitions at a moderate intensity of 50% of maximum repetition. As sessions progressed the number and intensity of exercises were gradually increased. 3. Cool down (10 min): stretching large muscle groups like hamstrings, hip flexors, shoulder muscles, etc. **Total days of training:** 16. **Duration of intervention:** 8 weeks. **Frequency per week:** 2 times. **Load adjustment:** not detailed in the text.
**Outcome measurements**	**1. Body composition:** BMI, fat mass in kg, fat mass in %, free fat mass in kg. (Bioelectrical impedance analysis). **2. Functional capacity:** 6MWT, chair stand test, postural stability test, abdominal strength, core flexor strength, modified push up test. **3. Quality of life:** OSQOL. **4. Surgical objectives:** not studied in the research.
**Global risk bias**	**Uncertain.**
**[11] Baillot et al. (2016)**	
**Randomization**	Used software stratified by sex and maximum aerobic capacity (> o ≤ 7 MET).
**Participants**	29 participants: 7 men—22 women. **Institution:** Centre hospitalier universitaire de Sherbrooke (CHUS), Quebec. **Country:** Canada. **Inclusion criteria:** candidates to BS, 18 years old or older, both sexes. **Exclusion criteria:** participants with comorbidities that prevent their participation in the prehabilitation program like a medical contraindication to practice physical activity, functional limitations that do not allow them to perform the 6MWT, not understanding the French language, or decompensated neuro-psychiatric pathology.
**Intervention**	**Gym program:** 1. Warm-up (10 min). 2. Aerobic phase: 30 min of exercise (treadmill, walking circuit, arm ergometer, elliptical machine) 3. Resistance phase: 20 to 30 min. 4. Cool down: 10 min. **Total days of training:** 24. **Intervention duration:** 12 weeks. **Frequency per week:** 2 times. **Load adjustment:** Aerobic: according to HRR from 55 to 75/80%. 8 levels were determined: A: 55%, B: 55%, C: 55%, D: 55%/65%, E: 65%, F: 65%/75%, G: 75% and H: 75%/85%. The duration was of 24 min at an A level and 30 min during rest. Resistance: increased from 2 to 3 sets, from 12 to 15 repetitions and at a weight of 5 to 12 lbs. for men, and 2 to 10 lbs. for women.
**Outcome measurements**	**1. Body composition:** BMI, fat mass in %. (Bioelectrical impedance analysis). **2. Functional capacity:** 6MWT, chair stand test, half squat test, arm curl test. **3. Quality of life:** WRQOL. **4. Surgical objectives:** not studied in the research.
**Global risk bias**	**High.**
**[15] Baillot et al. (2018)**	
**Randomization**	Used software stratified by sex and maximum aerobic capacity (> o ≤ 7 MET).
**Participants**	25 participants: 5 men—20 women. **Institution:** Centre hospitalier universitaire de Sherbrooke (CHUS), Quebec. **Country:** Canada. **Inclusion criteria:** candidates to a BS, 18-year-old or older, both sexes. **Exclusion criteria:** participants with comorbidities that prevent their participation in the prehabilitation program like a medical contraindication to practice physical activity, functional limitations that do not allow them to perform the 6MWT, not understanding the French language, or decompensated neuro-psychiatric pathology.
**Intervention**	**Gym program:** 1. Warm up: 10 min. 2. Aerobic phase: 30 min of exercise on the treadmill, walking circuit, arm ergometer, elliptical machine, aerobic dance. 3. Resistance phase: 20 to 30 min with small equipment, elastic bands, medicine balls, dumbbells, sticks. 4. Cool down: 10 min. **Total days of training:** 36. **Intervention duration:** 12 weeks. **Frequency per week:** 3 times. **Load adjustment:** Aerobic: according to a HRR from 55 to 75/80% (there are no more details in the article).
**Outcome measurements**	**1. Body composition:** BMI, free fat mass in %. (Bioelectrical impedance analysis). **2. Functional capacity:** 6MWT, half squat test. **3. Quality of life:** WRQOL. **4. Surgical objectives:** not studied in the research.
**Global risk bias**	**Uncertain.**
[16] **Funderburk et al. (2010)**	
**Randomization**	Unexplained.
**Participants**	7 participants: 1 man, 6 women. **Institution:** Hospital Pitt County Memorial, Rehabilitation center, Greenville. **Country:** United States of America. **Inclusion criteria:** candidates to a BS, 18 years old or older, both sexes. **Exclusion criteria:** no reports in the article.
**Intervention**	**Program of the institution:** The program included a warmup with exercises (walking in the water), strength and resistance exercises, and Ai Chi exercises for balance, core strengthening, and relaxation. Ai Chi is an aquatic exercise that was designed to increase relaxation, range of motion, and mobility. It is performed standing with the water at shoulder level using a combination of deep breathing and complete slow movements of the lower and superior extremities, as well as the torso. (There are no more details in the article). **Total days of training:** 24. **Intervention duration:** 12 weeks. **Frequency per week:** 2 times. **Load adjustment:** not detailed in the article.
**Outcome measurements**	**1. Body composition:** not studied in the article. **2. Functional capacity:** 6MWT, chair stand test, postural stability test, abdominal strength, core flexor strength, modified push up test. **3. Quality of life:** SF 36. **4. Surgical objectives:** not studied in the research.
**Global risk bias**	**Uncertain.**
[17] **Marcon et al. (2017)**	
**Randomization**	In blocks of 12 participants.
**Participants**	57 participants: 6 men—51 women. **Institution:** Hospital de Clinicas de Porto Alegre, Porto Alegre. **Country:** Brazil. **Inclusion criteria:** candidates to a BS, 18 years old or older, both sexes. **Exclusion criteria:** participants with comorbidities that prevent their participation in the prehabilitation program, participating in another supervised exercise program, patients with a class III or IV of heart functional capacity, orthopedic problems, severe retinopathy, severe neuropathy, drug addiction, severe mental illness, severe metabolic decompensation (250 mg/DI of blood glucose, systolic pressure over 200 mmHg, diastolic pressure over 100 mmHG).
**Intervention**	**Gym program:** Included aerobic exercise and stretching, intensity was measured by Borg’s scale, using a range between 2 to 4, considering it low to moderate intensity respectively. Arm and leg movements were alternated, moving to simulate walking. Stretching included: arms, legs, torso, and neck for 6 min after the aerobic phase in each session. (There are no more details in the article). **Total days of training:** 32. **Intervention duration:** 16 weeks. **Frequency per week:** 2 times. **Load adjustment:** not detailed in the article.
**Outcome measurements**	**1.- Body composition:** BMI. (Bioelectrical impedance analysis) **2.- Functional capacity:** 6MWT, VO_2_ max from equations after the test. **3.- Quality of life:** not studied in the research. **4.- Surgical objectives:** not studied in the research.
**Global risk bias**	**Uncertain**.

BS: bariatric surgery, BMI: body mass index, 6MWT: 6-min walk test, OSQOL: Obesity-Specific Quality of life, HRR: heart rate reserve, WRQOL: Weight-Related Quality of Life, SF 36: quality of life questionnaire related to health.

**Table 2 jcm-11-05091-t002:** Quality of evidence for the body composition, BMI, FM% and FFM Kg.

Certainty Assessment							No. of Patients		Effect		Certainty
No. of Studies	Study Design	Risk of Bias	Inconsistency	Indirectness	Imprecision	Other Considerations	Aerobic Physical Exercise, Resistance or Both	Standard Care (no Exercise)	Relative (95% CI)	Absolute (95% CI)	
4	randomised trials	Serious ^a^	Serious ^b^	not serious	Serious ^c^	none	54	61	-	SMD **0.71 SD fewer** (1.55 fewer to 0.12 more)	⨁◯◯◯ Very low
3	randomised trials	Serious ^d^	not serious	not serious	not serious	none	39	36	-	SMD **0.38 SD more** (0.47 fewer to 1.85 more)	⨁⨁⨁◯ Moderate
2	randomised trials	Serious ^e^	not serious	not serious	not serious	none	24	22	-	SMD **0.41 SD fewer** (1 fewer to 0.18 more)	⨁⨁⨁◯ Moderate

^a^ Downgraded one level due to risk of bias (>25% of the participants were from studies with a high risk of bias). ^b^ Downgraded one level due to clear inconsistency of results. ^c^ Downgraded one level due to imprecision. ^d^ Downgraded one level due to risk of bias (>25% of the participants were from studies with a high risk of bias). ^e^ Downgraded one level due to risk of bias (both studies with uclear risk of bias).

**Table 3 jcm-11-05091-t003:** Quality of evidence for the functional capacity of the 6MWT.

Certainty Assessment							No. of Patients		Effect		Certainty
No. of Studies	Study Design	Risk of Bias	Inconsistency	Indirectness	Imprecision	Other Considerations	Aerobic Physical Exercise, Resistance or Both	Standard Care (no Exercise)	Relative (95% CI)	Absolute (95% CI)	
2	randomised trials	not serious	not serious	not serious	not serious	none	33	28	-	SMD **2.59 SD more** (1.89 more to 3.3 more)	⨁⨁⨁⨁ High

**Table 4 jcm-11-05091-t004:** Quality of evidence for the quality of life total score.

Certainty Assessment							No. of Patients		Effect		Certainty
No. of Studies	Study Design	Risk of Bias	Inconsistency	Indirectness	Imprecision	Other Considerations	Aerobic Physical Exercise, Resistance or Both	Standard Care (no Exercise	Relative (95% CI)	Absolute (95% CI)	
3	randomised trials	serious^a^	not serious	not serious	not serious	none	28	25	-	SMD **0.88 SD more** (0.23 fewer to 1.99 more)	⨁⨁⨁◯ Moderate

## Data Availability

All data are available upon request to the corresponding author.

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
