# Peer review of "Effects of Supervised Physical Exercise as Prehabilitation on Body Composition, Functional Capacity and Quality of Life in Bariatric Surgery Candidates: A Systematic Review and Meta-Analysis"

_jcm, 2022, doi:10.3390/jcm11175091_

Round 1

Reviewer 1 Report

This systematic review is interesting but needs some major "adjustments"

1. Title: Prehabilitation is something more "comprehensive" than supervised physical exercise and for this reason the title should be modified

2. Introduction: in my opinion this should be re-written as it is too "pedantic": BS has already been established and well defined, there is no need to deep-dive into old classification (restrictive etc), as nowdays most effects are related to metabolic setpoint adjustments and complex gut-hormones, microbiota and bile acid interactions. ERAS again has a consolidated definitions and there is no need to report all well-known items

3. Forest plots Figs. 4, 5 and 6 report fovaours experimental before control while 7 and 8 control before experimental (any reason for that??

Author Response

Independent Review Report, Reviewer 1

Evaluation:

This systematic review is interesting but needs some major "adjustments"

  1. Title: Prehabilitation is something more "comprehensive" than supervised physical exercise and for this reason the title should be modified.
  2. Introduction: in my opinion this should be re-written as it is too "pedantic": BS has already been established and well defined, there is no need to deep-dive into old classification (restrictive etc.), as nowdays most effects are related to metabolic setpoint adjustments and complex gut-hormones, microbiota and bile acid interactions. ERAS again has a consolidated definitions and there is no need to report all well-known items.
  3. Forest plots Figs. 4, 5 and 6 report favors experimental before control while 7 and 8 control before experimental (any reason for that??

Reply:

Dear Reviewer 1 thank you very much for all your comments and suggestions. All the changes are highlighted in yellow in the revised manuscript. Please, find below our reply and indications where changes were made:

1.- The title was reformulated as “Effects of supervised physical exercise as prehabilitation on body composition, functional capacity and quality of life in bariatric surgery candidates: a systematic review and me-ta-analysis”, please check line 2 page 1.

2.- We appreciate the reviewer for this comment. We have reformulated the whole introduction. Please, check lines 39 to 72, page 1 and 2.

3.- Thank you for this observation, it was a mistake. The forest plots 7 and 8 were corrected to keep the consistency. Please, check Figures 7 (line 237 page 13) and 8 (line 247 page 14).

Reviewer 2 Report

The title of the manuscript includes surgical outcomes but in reality there is no data on it so I think it should be removed from the title. I think that duration of intervention of a week is ineffective. The benefits of exercise are well known and the manuscript does not appear to add relevant information. 

Author Response

Independent Review Report, Reviewer 2.

Evaluation:

1.- The title of the manuscript includes surgical outcomes but in reality there is no data on it so I think it should be removed from the title.

2.- I think that duration of intervention of a week is ineffective. The benefits of exercise are well known and the manuscript does not appear to add relevant information.

Reply:

Dear Reviewer 2 thank you very much for all your comments and suggestions. All the changes are highlighted in yellow in the revised manuscript. Please, find below our reply and indications where changes were made:

 1.- The title was reformulated as “Effects of supervised physical exercise as prehabilitation on body composition, functional capacity and quality of life in bariatric surgery candidates: a systematic review and me-ta-analysis”, please check  line 2 page 1.

2.- We agree with the reviewer that a week may not be effective in promoting improvements prior to surgery. However, this is not clear. In other areas, such as cardiothoracic surgery, previous studies (listed below) have evaluated the effect of physical exercise during prehabilitation prescribing a 1-week supervised exercise protocols and finding significant improvements before surgery. Therefore, we have considered studies that lasted one week in our literature search. However, the minimum intervention time frame of the studies included in this systematic review was of 8 weeks.

  1. Arbane, G., Douiri, A., Hart, N., Hopkinson, N.S., Singh, S., Speed, C., et al., 2014. Effect of postoperative physical training on activity after curative surgery for non-small cell lung cancer: a multicentre randomised controlled trial. Physiotherapy 100, 100–107.
  2. Pehlivan, E., Turna, A., Gurses, A., Gurses, H.N., 2011. The effects of preoperative short term intense physical therapy in lung cancer patients: a randomized controlled trial. Ann. Thorac. Cardiovasc. Surg. 17, 461–468.
  3. Lai, Y., Huang, J., Yang, M., Su, J., Liu, J., Che, G., 2017a. Seven-day intensive preoperative rehabilitation for elderly patients with lung cancer: a randomized controlled trial. J. Surg. Res. 209, 30–36.
  4. Lai, Y., Su, J., Qiu, P., Wang, M., Zhou, K., Tang, Y., et al., 2017b. Systematic short-term pulmonary rehabilitation before lung cancer lobectomy: a randomized trial. Interact. Cardiovasc. Thorac. Surg. 25, 476–483.
  5. Huang, J., Lai, Y., Zhou, X., Li, S., Su, J., Yang, M., et al., 2017. Short- term high-intensity rehabilitation in radically treated lung cancer: a three-armed randomized controlled trial. J. Thorac. Dis. 9, 1919–1929.)

  Reviewer 3 Report

The authors of the paper have chosen a topic that is currently highly interesting. The methodology used meets the requirements for this type of research, however some modifications/clarifications could improve the quality of the final version. For example, the confusing introductory part can be the information about selection criteria, as only randomized clinical trials that examined the effectiveness of supervised physical exercise as prehabilitation were included where the main expected outcomes were body composition, functional capacity, quality of life, and surgical outcomes. It is to be expected that no RCTs are included as none of the screened ones reported surgical outcomes. Reformulation or adding information that this criterion was not considered because it was not studied for any RCT will good. This is partially stated in the limitations. The risk of assessment (line 135) says there are 6 bias domains, but there are only 5 mentioned. Consider whether Figure 2 is necessary because Figure 3 shows similar information. Also, translate all parts of the header row into English (in tables 2-4).

Author Response

Independent Review Report, Reviewer 3.

Evaluation:

1.- The authors of the paper have chosen a topic that is currently highly interesting. The methodology used meets the requirements for this type of research, however some modifications/clarifications could improve the quality of the final version. For example, the confusing introductory part can be the information about selection criteria, as only randomized clinical trials that examined the effectiveness of supervised physical exercise as prehabilitation were included where the main expected outcomes were body composition, functional capacity, quality of life, and surgical outcomes. It is to be expected that no RCTs are included as none of the screened ones reported surgical outcomes. Reformulation or adding information that this criterion was not considered because it was not studied for any RCT will good. This is partially stated in the limitations.

2.- The risk of assessment (line 135) says there are 6 bias domains, but there are only 5 mentioned.

3.- Consider whether Figure 2 is necessary because Figure 3 shows similar information.

4.- Also, translate all parts of the header row into English (in tables 2-4).

Reply:

Dear Reviewer 3 thank you very much for all your comments and suggestions. All the changes are highlighted in yellow in the revised manuscript. Please, find below our reply and indications where changes were made:

 1.- We agree with the reviewer. This paragraph was reformulated to improve clarity. Please, check section 2.5 (lines 101 to 111 page 3).

2.- The missing risk assessment (attrition) was added. Please, check line 133, page 3.

3.- Figure 2 presents a summary for all the studies, while figure 3 shows the result of the bias analysis for each study individually. The authors consider that both figures could be kept, but we are opened to remove it in case of editor and reviewer consider it is necessary.

4.- We are sorry for the mistake. The parts of the header are now translated. Please check table 2, 3 and 4, pages 12, 13 and 14.

  Reviewer 4 Report

Dear Authors,

you have to be commended for this important piece of work.

I have just one suggestion: to gather in more evidence in the discussion which are the methods implied for improved outcomes. I mean, to include in the discussion a description of the protocols adopted in the articles you selected

Author Response

Evaluation:

Dear Authors, you have to be commended for this important piece of work.

I have just one suggestion: to gather in more evidence in the discussion which are the methods implied for improved outcomes. I mean, to include in the discussion a description of the protocols adopted in the articles you selected

Reply:

Dear Reviewer 4 thank you very much for all your comments and suggestions. All the changes are highlighted in yellow in the revised manuscript. Please, find below our reply and indications where changes were made:

 The observation is greatly appreciated. A paragraph was added where the protocol of the included studies are discussed in more details. Please, check page 15, lines 272 to 287.

Round 2

Reviewer 2 Report

The paper can be accepted in this revised form.